# Improve Certified Training with Signal-to-Noise Ratio Loss to Decrease Neuron Variance and Increase Neuron Stability

**Tianhao Wei**                                          *twei2@andrew.cmu.edu*
*Robotics Institute, Carnegie Mellon University*

**Ziwei Wang**                                           *ziweiwa2@andrew.cmu.edu*
*Robotics Institute, Carnegie Mellon University*

**Peizhi Niu**                                           *pniu@andrew.cmu.edu*
*Robotics Institute, Carnegie Mellon University*

**Abulikemu Abuduweili**                                 *abulikea@andrew.cmu.edu*
*Robotics Institute, Carnegie Mellon University*

**Weiye Zhao**                                           *weiyezha@andrew.cmu.edu*
*Robotics Institute, Carnegie Mellon University*

**Casidhe Hutchison**                                    *fhutchin@nrec.ri.cmu.edu*
*Robotics Institute, Carnegie Mellon University*

**Eric Sample**                                          *esample@nrec.ri.cmu.edu*
*Robotics Institute, Carnegie Mellon University*

**Changliu Liu**                                         *cliu6@andrew.cmu.edu*
*Robotics Institute, Carnegie Mellon University*

**Reviewed on OpenReview:** *https://openreview.net/forum?id=iVOjktFZ5Y*

## Abstract

Neural network robustness is a major concern in safety-critical applications. Certified robustness provides a reliable lower bound on worst-case robustness, and certified training methods have been developed to enhance it. However, certified training methods often suffer from over-regularization, leading to lower certified robustness. This work addresses this issue by introducing the concepts of neuron variance and neuron stability, examining their impact on over-regularization and model robustness. To tackle the problem, we extend the Signal-to-Noise Ratio (SNR) into the realm of model robustness, offering a novel perspective and developing SNR-inspired losses aimed at optimizing neuron variance and stability to mitigate over-regularization. Through both empirical and theoretical analysis, our SNR-based approach demonstrates superior performance over existing methods on the MNIST and CIFAR-10 datasets. In addition, our exploration of adversarial training uncovers a beneficial correlation between neuron variance and adversarial robustness, leading to an optimized balance between standard and robust accuracy that outperforms baseline methods.

## 1 Introduction

Ensuring robustness in neural networks, especially those deployed in safety-critical systems, is paramount. Recent efforts have led to the development of various algorithms to both empirically (Madry et al., 2017; Goodfellow et al., 2014) and formally (Liu et al., 2021) assess the robustness of neural networks, giving

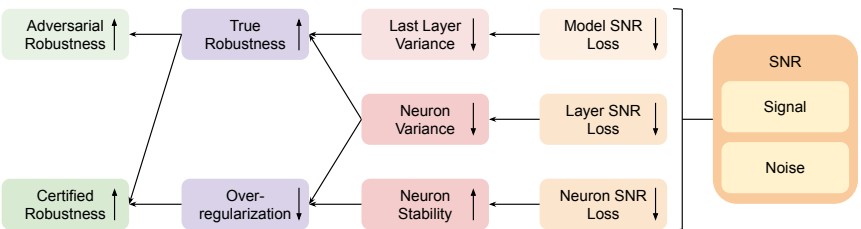

Figure 1: Overview of the analysis and the method. The Signal-to-Noise Ratio inspired losses at different scale improve neuron variance and neuron stability, leading to higher true robustness and less over-approximation, and consequently higher certified robustness and higher adversarial robustness.

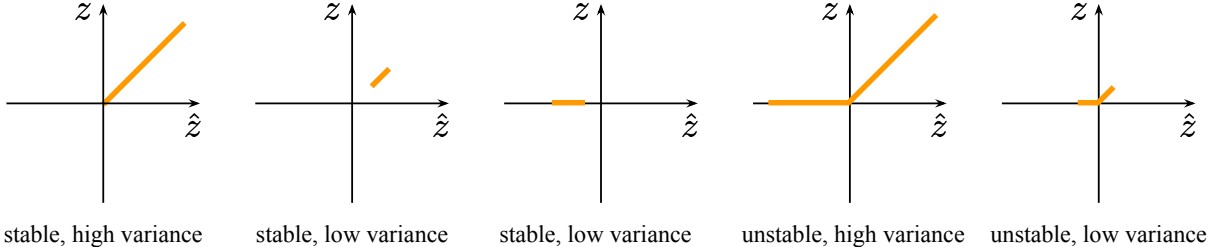

| stable, high variance | stable, low variance | stable, low variance | unstable, high variance | unstable, low variance |

Figure 2: Neuron variance and stability in ReLU activation. $\hat{z}$ denotes the input range of the ReLU, $z$ denotes the output range of the ReLU. A neuron is unstable if its input crosses 0.

rise to the concepts of adversarial and certified robustness. While certified robustness offers a reliable lower bound on the worst case output robustness (termed as true robustness), adversarial robustness provides a near upper bound. However, computationally, certified robustness is usually much lower than adversarial robustness because of the difficulty of certification (Liu et al., 2021). To address this problem, certified training methods were proposed (Gowal et al., 2018; Mao et al., 2023b) to make the model easier to certify within finite time. However, a model being easier to certify does not necessarily equate to higher certified robustness. De Bartolomeis et al. (2023) have shown that a major challenge of certified training is *over-regularization*, which excessively penalizes the model to reduce the difficulty of certification, leading to lower (although faster to compute) certified robustness than adversarially trained models (Mao et al., 2023a).

In this work, we aim to bridge this gap by enhancing certified training methods. We focus on reducing the over-regularization to achieve higher certified robustness. In general, over-regularization is caused by over-approximation of the worst case output of the neural network. In practical certification algorithms, this over-approximation comes from convex relaxation of the neural network (Liu et al., 2021; Müller et al., 2022) which overcomes the nonlinearity of neural networks for easier certification. Many convex relaxation methods were proposed, some are tight but slow (Tran et al., 2021) while some are loose but fast (Gowal et al., 2018). It is a common belief that the certified robustness is generally hindered by the number of unstable neurons (i.e., neurons introducing nonlinearity, hence requiring convex relaxation) (Shi et al., 2021; De Bartolomeis et al., 2023). However, a detailed quantification of how the collective behavior of single neurons impacts certified robustness remains absent from the literature.

To analyze these relationships, we first introduce two concepts *neuron variance* and *neuron stability* to characterize behaviors of neurons. Neuron variance is the variance of a neuron output given an input range of the model. As shown in fig. 2, we say a neuron has high variance if the output varies a lot under perturbation. Neuron stability is a binary status defined for piece-wise linear activations such as ReLU and Leaky ReLU. We call a neuron unstable if it introduces non-linearity, and otherwise stable. For example, for ReLU activation, if the input of the ReLU is always positive or always negative, then the neuron is considered stable.

We use both empirical and theoretical methods to analyze how over-approximation relates to neuron variance and neuron stability. In the empirical study, to isolate the impacts of neuron variance and neuron stability on over-approximation, we shift bias terms to control the number of unstable neurons and neuron variance. It is revealed that the over-approximation is positively related to neuron stability under tight convex relaxations, such as Star (Tran et al., 2021) and Zonotope (Singh et al., 2018), and the over-approximation is positively related to the neuron variance for both tight relaxation and loose relaxation, such as the commonly used box relaxation (also known as IBP) (Gowal et al., 2018). Since the best practice of certified training is to train with box relaxation and verify with tight relaxation (Jovanović et al., 2021), it is essential to both reduce the number of unstable neurons and reduce neuron variance. However, directly using neuron variance and neuron stability as loss functions is impractical due to the binary nature of neuron stability and the varying scales of neuron variance across different layers.

To tackle this challenge, we introduce the concept of Signal-to-Noise Ratio (SNR), offering a novel perspective on model robustness and providing two surrogate losses for neuron variance and neuron stability. SNR is a widely used metric in signal processing but has not been used to analyze neural network robustness so far. By viewing the neuron-by-neuron output of the clean input as the signal and the output deviation under perturbation as noise, our intuition is that a robust model should aim to suppress the noise and maximize the SNR. The de facto loss used in adversarial training, TRADES (Zhang et al., 2019a), can be viewed as a model level SNR loss that only applies to the last layer of the model. However, unlike adversarial training where only the robustness of the last layer matters, in certified training, the robustness of intermediate layers plays a vital role because the convex relaxation error accumulates through the network. Therefore, it is essential to optimize these intermediate layers to improve neuron variance and stability for reducing over-approximation. We define the inverse of the layer-level SNR as the loss for neuron variance, which is exactly a weighted sum of neuron variance based on the layer-wise output scale. We also define a continuous surrogate loss for neuron stability based on the neuron-level SNR. The overview of our analysis and method is shown in fig. 1. Our experiment results show that the SNR inspired losses effectively improve the certified robustness, and outperform state-of-the-art methods on MNIST and CIFAR-10.

Furthermore, given the fact that SNR losses improve certified robustness, we hypothesize that SNR losses could also improve adversarial robustness or even true robustness. To validate the hypothesis, we first draw the layer-level SNR across network layers of adversarial-trained versus standard-trained models to show the correlation between neuron variance and the adversarial robustness, where we observe a distinctive behavior. Notably, the layer-level SNR has an increasing trend over depth in adversarial-trained models, contrary to the consistency seen across all layers in standard models, which demonstrates a strong correlation between neuron variance and the adversarial robustness. Then we combine the variance loss with two popular adversarial training methods, TRADES (Zhang et al., 2019a) and AD (Madry et al., 2017), which can be viewed as model-level SNR. Our experiments manifest the efficacy of this regularization by achieving a higher robust accuracy while maintaining the same natural accuracy on MNIST and CIFAR-10, demonstrating that the variance loss effectively improves the adversarial robustness. Since adversarial robustness is a close approximation of the true robustness (Müller et al., 2022), this finding suggests that neuron variance is strongly correlated with true robustness (ideally, an extremely robust model should have zero neuron variance).

In summary, our contributions are multifaceted: 1. we numerically and analytically demonstrate that the reachable set over-approximation has a positive correlation with both neuron stability and neuron variance. 2. We extend the concept of Signal-to-Noise Ratio (SNR) to model robustness, and design two losses based on it to reduce the neuron variance and the number of unstable neurons. 3. We apply these two losses to certified training and demonstrate their efficacy by outperforming state-of-the-art methods on MNIST and CIFAR-10. 4. We show that the true robustness of a model is correlated with neuron variance and can be improved by variance loss through adversarial training, achieving a better trade-off between robust accuracy and standard accuracy on MNIST and CIFAR-10.

## 2 Background

**Training robust neural networks** can generally be viewed as solving the following min-max optimization:

$$\min_\theta \mathbb{E}_{(\mathbf{x},y)\in\mathcal{X}} \left[ \max_{\delta\in\Delta(\mathbf{x})} \mathcal{L}\left(f_\theta(\mathbf{x}+\delta), y\right) \right] \tag{1}$$

where $f_\theta$ denotes a neural network parameterized by $\theta$, $\mathcal{X}$ is the data distribution, $\mathbf{x}$ is a data sample, $y$ is its ground-truth label, $\delta$ is a perturbation constrained in the set $\Delta(\mathbf{x})$, and $\mathcal{L}$ is the loss function.

**Adversarial training methods** solve the inner maximization in equation 1 with adversarial attack, and solve the outer minimization with normal training with corrected labels (Goodfellow et al., 2014). Madry et al. (2017) further advanced adversarial training using a strong iterative adversary, demonstrating its capability to train models that are highly robust to many adversarial attacks. However, it's worth noting that adversarial training can compromise the generalization performance of deep learning models (Zhang et al., 2019a). Consequently, researchers have introduced variations of adversarial training aiming to strike a balance between accuracy and robustness, as exemplified by methods such as TRADES (Zhang et al., 2019a), Interpolated Adversarial Training (Lamb et al., 2019), and other approaches (Raghunathan et al., 2020; Han et al., 2020). In this work, we show that a better trade-off on some datasets can be achieved by combining our method with TRADES (Zhang et al., 2019a) or AD (Madry et al., 2017). Furthermore, TRADES and AD can be viewed as an SNR loss with a focus on the SNR of the last layer of the model.

**Certified training methods** optimize an upper bound of the inner maximization in equation 1. The upper bound is derived from bound propagation (Mao et al., 2023b). There are two classes of methods to derive the upper bound: 1. *Sound* methods provide a strict upper bound, such as IBP (Gowal et al., 2018), Star (Tran et al., 2021), and DeepZ (Singh et al., 2018), but the upper bound is generally loose because of over-approximation; 2. *Unsound* methods, attempting to reduce over-approximation, provide more accurate estimations of the upper bound at the cost of soundness Müller et al. (2022); Mao et al. (2023a).

**Over-regularization** is a common problem in certified training. Because certified training methods rely on convex relaxation to derive the upper bound, over-approximation of the upper bound often exists (Mao et al., 2023a). There is a consistent empirical understanding of the relative tightness of different convex relaxations (Jovanović et al., 2021). However, a tight relaxation, even though it leads to less over-regularization, does not necessarily produce higher certified accuracy in certified training due to the continuity and sensitivity of the training dynamics (Jovanović et al., 2021). Therefore, new methods to address over-regularization are needed. We propose to reduce over-approximation by reducing unstable neurons, which is orthogonal to existing convex relaxation based methods.

## 3 How Do Neuron Variance and Stability Affect Reachable Set Over-approximation?

In this section, we present a new analysis to show that both neuron variance and stability affect over-regularization through the introduction of reachable set over-approximation, which motivates our method in section 4. The limitations of existing analysis, which prevents the detailed quantification of the cause of over-approximation, will be first discussed with a case study. To address the limitations, we present a new analysis, controlling neuron variance and stability by shifting bias terms. A numerical study and an analytical study are developed based on this method, showing that the over-approximation in Box relaxation is dominated by neuron variance, and that the over-approximations in tight relaxations, such as Star and Zonotope, is dominated by neuron stability. Therefore, it is important to minimize both of them to achieve higher model robustness.

We study the over-approximation in certification with reachability based methods Liu et al. (2021), where a reachable set $Y$ of the output is computed. Directly comparing the reachable set gives us an intuition of the worst case over-approximation in arbitrary directions. In contrast, other methods focus on checking whether the output specification is violated or not, hence not computing $Y$.

### 3.1 Limitation and Solution

It is well known that an unstable neuron introduces over-approximation at the neuron level (Zhang et al., 2018). Consequently, a common belief is that the performance of certification reduces with the growth of the number of unstable neurons because they result in reachable set over-approximation (Xu et al., 2020; Lee et al., 2021). However, this belief has not been directly and rigorously verified because it is difficult to control the number of unstable neurons without affecting the true reachable set. Existing approaches control the number of unstable neurons indirectly through factors such as model width, depth, and perturbation budget (De Bartolomeis et al., 2023; Mao et al., 2023b). However, as is to be shown in fig. 3, these factors also change the true reachable set, making it challenging to quantify the over-approximation.

**Case study** We consider fully connected neural networks with 2-dimensional outputs for visualizing the last layer reachable set. We compute the exact reachable set by enumerating all linear pieces of the neural network. We consider three popular convex relaxation: Box (also known as IBP), Zonotope, and Star. Box keeps dimension-wise lower bounds and upper bounds. Zonotope is defined as $\{x \mid x = c + \sum_i \alpha_i g_i, \text{ s.t.} |\alpha| < 1\}$, with $c$ and $g_i$ as layer-output-dimensional vectors. The vector $c$ denotes the center of the set; $g_i$ is a generator of the set; and $\alpha$ is a scalar. In short, Zonotope contains all points that can be represented as the center plus a weighted sum of the generators, where the $L_1$ norm of the weight should be less than 1. Star uses a similar representation as Zonotope, but with a different constraint on the weight: $\{x \mid x = c + \sum_i \alpha_i g_i, \text{ s.t.} A\alpha < b\}$. This constraint $A\alpha < b$ is more flexible than $|\alpha| < 1$, therefore Star is tighter than Zonotope but is slower to compute. We also draw the convex hull of the exact reachable set to illustrate the optimal convex relaxation. As shown in fig. 3, when these factors change, both the true reachable set and the convex relaxation change. The 3-4-4-2 model is randomly initialized. To control the variables, other models are obtained by directly taking the subset of the 3-4-4-2 model parameters. For example, 3-4-2 is obtained by removing the third layer of 3-4-4-2. Figure 3a and fig. 3b demonstrate the depth change. Figure 3b, fig. 3c and fig. 3f demonstrate the width change. Figure 3b and fig. 3d show the input set change. The input set is reduced by half on each dimension in fig. 3d (the volume is 1/8 of the original input). Because both the true reachable set and the convex relaxation change, we cannot directly conclude that the over-approximation is determined by the number of unstable neurons.

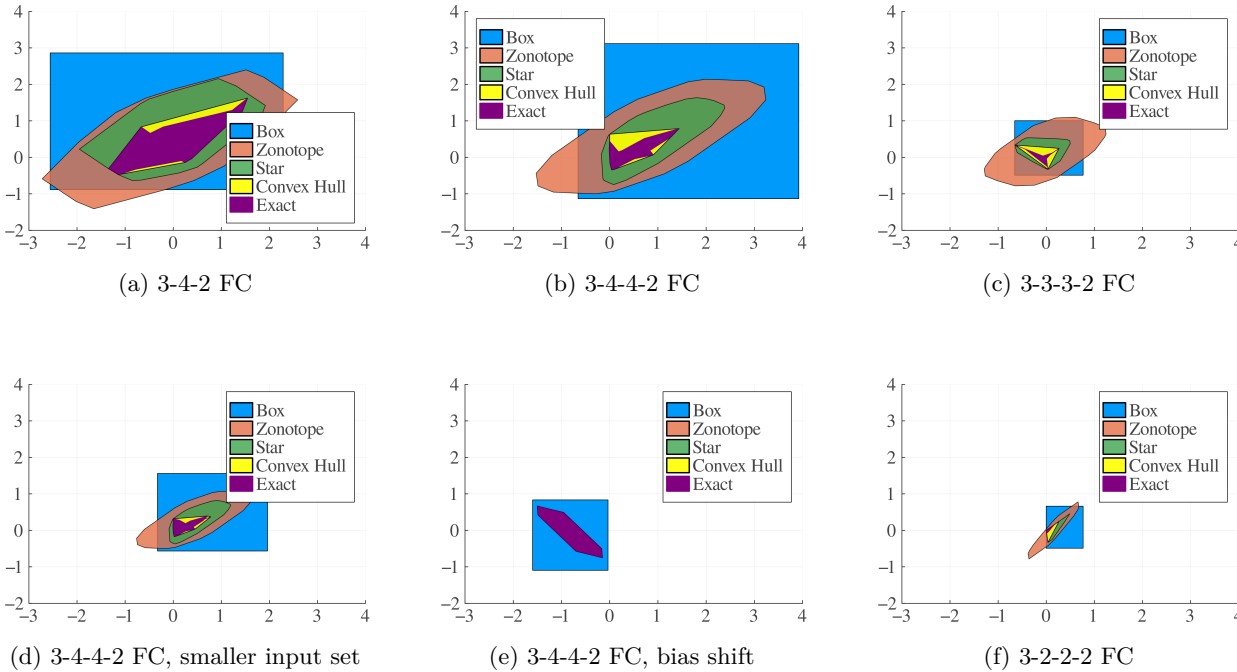

Figure 3: How do different factors affect the true reachable set and its convex relaxation for models with 2 dimensional outputs. a-b-c-d FC denotes a fully connected model with four layers of the size a, b, c, and d.

**Solution**  To address this problem, we control the number of unstable neurons by shifting the bias before each activation function. As shown in fig. 3e, this method can keep the exact reachable set at a similar scale while reducing the over-approximation. Given the lower bound and upper bound of each neuron in the model, we can make an unstable neuron stable by subtracting either its upper bound or its lower bound, making it always negative (inactive) or positive (active). To get an accurate estimation of the lower bound and upper bound of each neuron, a sampling based method is adopted. We sample multiple inputs from the input set and perform inference on the model. The extreme values of each neuron (adding a constant margin) are used as their lower bound and upper bound.

To explain why bias shift is a better way to control the number of unstable neurons than width, depth and input set size, we can consider the deep linear network (DLN) $f(x) = \Pi_{i=1}^{l} W_i \, x$, which describes ReLU networks for infinitesimal perturbation magnitudes. Because DLN retains the layer-wise structure and joint non-convexity in the weights of different layers, it is a popular analysis tool (Mao et al., 2023b; Ribeiro et al., 2016). The reachable set of DLN is an affine transformation of the input set. It is obvious that changing width, depth and the input set changes the reachable set size. However, shifting bias results in only a translated reachable set for a DLN because $f'(x) = W_l(W_{l-1}(\cdots(W_1x+b_1)+b_2)+\cdots+b_{l-1}) = \Pi_{i=1}^{l} W_i \, x + c$ has the same transformation as $f(x)$ except for a different constant.

This analysis does not directly generalize to ReLU networks because bias shift actually changes the exact reachable set when ReLU is present. As shown in fig. 4, for a neuron, the reachable set becomes larger after shifting it to positive, and becomes smaller after shifting it to negative. However, we can still isolate the source of over-approximation by controlling variables. We can study bias shift for ReLU in three cases: 1. shift all unstable neurons to inactive: as the unstable neurons reduce, the exact reachable set shrinks while the non-linearity decreases. 2. shift all unstable neurons to active: as the unstable neurons reduce, the exact reachable set increases while the non-linearity decreases. 3. shift unstable neurons to active or inactive, a random half of the unstable neurons are shifted to active and the other half are shifted to inactive. As the unstable neurons reduce, the nonlinearity reduces. However, empirical observations indicate that the exact reachable set remains approximately the same in scale, as shown in fig. 3e. By comparing the over-approximation in these three cases, we can identify the dominating source of over-approximation for different solvers.

## 3.2 Numerical study

We conducted the numerical test on a randomly generated 4-layer fully connected neural network, with 50 neurons per layer. Because it is difficult to directly compare high dimensional polytopes, we measure the size of the reachable set by the maximal dimension-wise radius. The radius of the exact set is estimated by the weight matrix norm of the corresponding DLN as used by Mao et al. (2023b). As shown in fig. 5, Box relaxation grows when neurons shift to active, and reduces when neuron shift to inactive, even though in both cases, the number of unstable neurons reduces. This reveals that the over-approximation of the box relaxation is dominated by the dimensionwise reachable set (neuron variance), which is expected (Gowal et al., 2018). But in contrast, the performance of tight convex relaxation methods such as Star and Zonotope are dominated by the nonlinearity (number of unstable neurons). Even though in the shift to active case, the exact reachable set grows larger when there are fewer unstable neurons, Star and Zonotope still achieve smaller reachable sets, indicating less over-approximation. This demonstrates that the number of unstable neurons plays a vital role in over-approximation for tight relaxations.

It has been shown that the most effective way for certified training is to train the network with Box relaxation followed by verifying the network with tight relaxation such as Star and Zonotope Jovanović et al. (2021). Therefore, it is important to reduce the over-approximation of both methods, which motivates us to reduce the number of unstable neurons and the dimension-wise reachable set, thus reducing neuron variance.

## 3.3 Analytical study

To confirm that neuron stability is important for tight solvers and the neuron variance is important for all solvers, we give an analytical solution of the over-approximation in the shift to active case. We consider a two layer neuron network with two dimensional input and output, which can be denoted by $\hat{z} = W_1 x + b_1$, $z =$

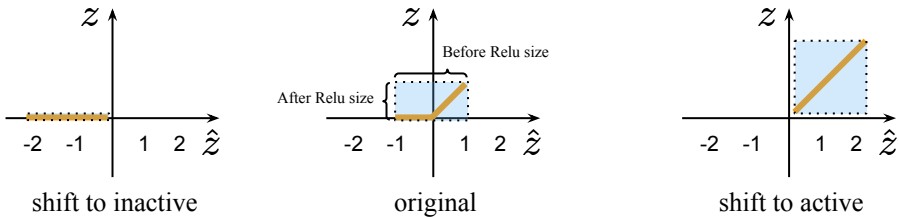

Figure 4: One dimensional reachable set after bias shift. Shift to active leads to dimension-wise larger reachable set, and shift to inactive leads to dimension-wise smaller reachable set.

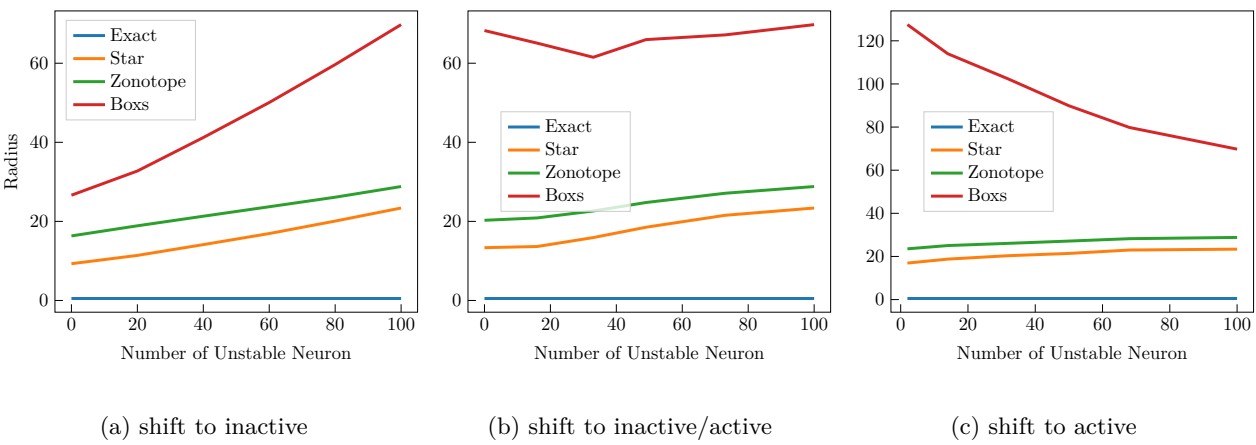

Figure 5: Level of over-approximation against neuron stability under three bias shift cases. The over-approximation (the gap between the convex relaxation radius and the estimated exact radius) grows with the number of unstable neurons consistently for tight solvers such as Star and Zonotope. But the over-approximation of Box is dominated by the reachable set radius (controlled by the bias shift method).

$\sigma(\hat{z})$, $y = W_2 z + b_2$, where $x$ is the input, $\hat{z}$ is the pre-activation layer, $z$ is the hidden layer, and $y$ is the output. We denote the reachable set of each layer by $X$, $\hat{Z}$, $Z$, $Y$ respectively. In general, it is difficult to give an analytical solution of the Star over-approximation because it involves solving a linear programming, which typically requires numerical solvers. Therefore, we consider a special case where $\hat{Z}$ is an $L_1$-norm ball and $W_2$ is a rotation matrix as shown in fig. 6. Specifically, $W_2$ rotates $z$ by $\pi/4$. We denote the $L_1$ norm ball by $(c_x, c_y)$ and $r$, representing the center and radius respectively. We measure the over-approximation by the after rotation $Y$-radius of the reachable set.

We analyze the over-approximation in the shift to active case under different number of unstable neurons: 2 unstable neurons, 1 unstable neuron, and 0 unstable neuron. Without loss of generality, we assume $c_x < c_y$ in all cases. The detailed derivation of all cases is in appendix A. A summary of the analytical solutions is in table 1. When there are fewer unstable neurons, Box over-approximation increases while Star over-approximation decreases, showcasing that stability is important for tight solvers such as Star. Both Box over-approximation and Star over-approximation are related to neuron variance, represented by the radius $r$ of the $L_1$-ball, indicating that the variance is a critical factor for all solvers. The analytical solution confirms that enhancing neuron stability and minimizing neuron variance are critical in reducing over-approximation (and consequently in improving certified robustness) as we observed in the numerical test.

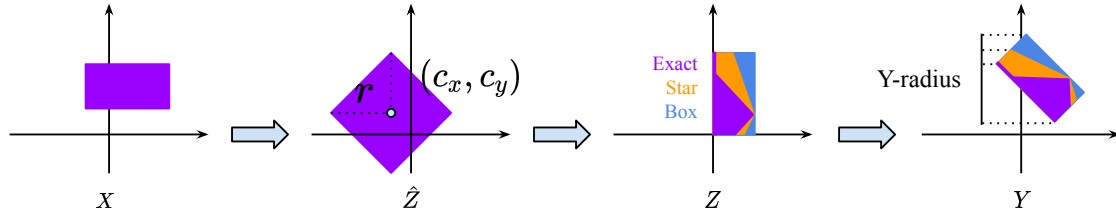

Figure 6: Reachable set propagation when $\hat{Z}$ is an $L_1$ norm ball, and $W_2$ is a $\pi/4$ rotation matrix. In this example, there are two unstable neuron when going through the ReLU activation. We analyze the over-approximation by computing the Y-radius of output set.

| Y-radius of | 2 unstable | 1 unstable | 0 unstable |
|---|---|---|---|
| Exact | $\frac{\sqrt{2}}{2}(c_y + r + \max\{0, c_x\})$ | $\sqrt{2}r$ | $\sqrt{2}r$ |
| Box | $\frac{\sqrt{2}}{2}(c_x + c_y + 2r)$ | $\frac{\sqrt{2}}{2}(c_x + 3r)$ | $2\sqrt{2}r$ |
| Star | $\frac{\sqrt{2}}{2}(\frac{1}{2}c_x + c_y + \frac{3}{2}r)$ | $\frac{\sqrt{2}}{2}(\frac{1}{2}c_x + \frac{5}{2}r - \max\{c_x, 0\})$ | $\sqrt{2}r$ |
| Box - Exact | $\frac{\sqrt{2}}{2}(r + \min\{c_x, 0\}) \in [\mathbf{0}, \frac{\sqrt{2}}{2}\mathbf{r}]$ | $\frac{\sqrt{2}}{2}(r + c_x) \in [\mathbf{0}, \sqrt{2}\mathbf{r}]$ | $\sqrt{2}\mathbf{r}$ |
| Star - Exact | $\frac{\sqrt{2}}{4}(r - |c_x|) \in [\mathbf{0}, \frac{\sqrt{2}}{4}\mathbf{r}]$ | $\frac{\sqrt{2}}{4}(r - |c_x|) \in [\mathbf{0}, \frac{\sqrt{2}}{4}\mathbf{r}]$ | $\mathbf{0}$ |

Table 1: Analytical solution of Y-radius over-approximation for the given example. Box - Exact and Star - Exact measure the Box and Star over-approximation respectively. $(c_x, c_y)$ and $r$ denote the center and radius of the $L_1$ norm ball.

# 4 Method: SNR loss

We have shown that both the number of unstable neurons and neuron variance are vital in reducing over-approximation. In this section, we propose two losses: variance loss and stability loss, to reduce neuron variance and enhance neuron stability respectively. We observe a strong connection between model robustness and the concept of Signal-to-Noise Ratio (SNR), widely recognized in signal processing and communications. We extend SNR to model robustness and define layer-level SNR and neuron-level SNR, which inspires the variance loss and stability loss. The layer-level SNR loss can be viewed as a weighted sum of neuron variance based on the output range of each layer, and the neuron-level SNR loss can be viewed as a weighted sum of neuron noises based on a neuron's distance to the stability boundary (when signal equals noise).

## 4.1 Mathematical Definition of Signal-to-Noise Ratio

We denote a neural network by $f : X \mapsto Y$, the output of $i$-th layer by $f^i(\boldsymbol{x})$, and the output of the $i$-th layer $j$-th neuron by $f^i_j(x)$. Considering an original input $\boldsymbol{x}_0$ and $m$ perturbed inputs $\boldsymbol{x}_1, \cdots, \boldsymbol{x}_m$, where $\boldsymbol{x}_i \in \Delta(\boldsymbol{x}_0) \subseteq X$, we define the signal of a neuron as $s^i_j = f^i_j(\boldsymbol{x}_0)$, and the noise of a neuron as $n^i_j = f^i_j(\boldsymbol{x}_k) - f^i_j(\boldsymbol{x}_0)$. The intuition is that the signal is the truly useful output, and the noise is the output deviation caused by perturbations. The max noise can be viewed as an under-approximation of the size of the dimension-wise reachable set. $\boldsymbol{x}_i$ can be sampled from an arbitrary distribution or adversarially searched. But adversarial search usually gives near-worst-case noises and therefore gives a better under-approximation of the reachable set. Similarly, we define the squared signal of a layer as $S^2_i := \sum_j {s^i_j}^2$, and the squared noise of a layer as $N^2_i := \frac{1}{m}\sum_j \sum_{k=1}^m {n^i_{jk}}^2$. The SNR of a layer is defined as

$$\text{SNR}^i := \frac{\sum_j {s^i_j}^2}{\frac{1}{m}\sum_j \sum_{k=1}^m {n^i_{jk}}^2} = \frac{\|f^i(\boldsymbol{x}_0)\|_2^2}{\frac{1}{m}\sum_{k=1}^m \|f^i(\boldsymbol{x}_k) - f^i(\boldsymbol{x}_0)\|_2^2}. \quad (2)$$

## 4.2 Layer SNR: Variance loss

We can see the denominator of the layer SNR is exactly the neuron variance and the numerator is the scale of the layer output. Therefore, we can directly use the inverse of SNR as the loss to minimize neuron variance.

$$\mathcal{L}^i_{var} := \frac{\frac{1}{m}\sum_{k=1}^m \sum_{j=1}^{d_i} {n^i_j}^2}{\sum_{j=1}^{d_i} {s^i_j}^2} = \frac{\frac{1}{m}\sum_{k=1}^m \|f^i(\boldsymbol{x}_k) - f^i(\boldsymbol{x}_0)\|_2^2}{\|f^i(\boldsymbol{x}_0)\|_2^2}. \tag{3}$$

During experiments, we found that the model tends to minimize the loss by enlarging the signal through increasing the weight scale instead of reducing the noise. We hypothesize that this due to the loss function causing the model to enlarge the signal instead of reducing the noise. Reducing noise requires the model to learn a truly robust representation of semantic features, whereas enlarging the signal only requires increasing the weight scale. Therefore, we exclude the gradient w.r.t the signal, fi(x0), to prevent the model from cheating and only increasing the signal. So, we only optimize the loss to reduce noise, not to enlarge signal. The term $1/\|f^i(\boldsymbol{x}_0)\|_2^2$ serves as a loss weight to balance the penalty of different layers.

## 4.3 Neuron SNR: Stability loss

A neuron is unstable if $\max_p f^i_j(\boldsymbol{x}_p) > 0$ and $\min_q f^i_j(\boldsymbol{x}_q) < 0$. We design the following stability loss to reduce unstable neurons:

$$\mathcal{L}^i_{stb} := \frac{1}{m}\sum_{k=1}^m \sum_{j=1}^{d_i} \frac{|n^i_j|}{{s^i_j}^2 + \epsilon} = \frac{1}{m}\sum_{k=1}^m \sum_{j=1}^{d_i} \frac{|f^i_j(\boldsymbol{x}_k) - f^i_j(\boldsymbol{x}_0)|}{{f^i_j}^2(\boldsymbol{x}_0) + \epsilon}. \tag{4}$$

The intuition behind this loss is to encourage the noise to be small when the signal is close to zero, such that unstable neurons can be reduced. We also allow the signal to shift away from zero to reduce the number of unstable neurons. In contrast to the layer-wise variance loss, the stability loss is a neuron-wise loss. Since the loss does not decrease by simply increasing the weight matrix scale, we permit gradient with respect to the signal. We use $L_1$ norm of $n^i_j$ because we are most concerned with the number of unstable neurons instead of the maximal noise. We use ${s^i_j}^2 + \epsilon$ instead of $s^i_j$ in the denominator to avoid $n^i_j/s^i_j \to \infty$ when $s^i_j \to 0$ and to make the gradient of $\mathcal{L}^i_{stb}$ continuous and smooth.

# 5 Experiments

In this section, we show that the propose two losses $\mathcal{L}_{var^i}$ and $\mathcal{L}_{stb^i}$ can be used to improve model robustness in both certified training and adversarial training.

## 5.1 Certified Training

**Datasets and Implementation Details** We first introduce the datasets on which we conducted experiments and the implementation details of our method. The datasets include MNIST LeCun et al. (2010) and CIFAR-10 Krizhevsky et al. (2009). Following the experimental settings in SABR Müller et al. (2022), we adopted horizontal flip and random cropping with normalization for data augmentation in CIFAR-10. We used the test sets of MNIST and CIFAR-10 for evaluation, where the metrics include natural accuracy and certified accuracy. The former demonstrates the performance on clean samples, and the latter depicts the ratio of samples that are guaranteed to be correctly classified under any perturbation within a given magnitude. The perturbation magnitudes for experiments on MNIST were set to 0.1 and 0.3, respectively, and those on CIFAR-10 were assigned to 2/255 and 8/255. We employed MN-BAB Ferrari et al. (2022) for certification, where the time limit for verifying a single input was 500 and 1000 seconds for samples in MNIST and CIFAR-10, respectively. The training and evaluation is carried on a single 16GB NVIDIA RTX A4000 GPU.

The model we utilize is CNN7, following the settings described in Müller et al. (2022). We will leave the study on more complex models for future work, as they require more advanced verification algorithms to

Table 2: Comparison of the natural accuracy and the certified accuracy with different perturabation magitutes and on different datasets. The verification method is MN-BAB and the time limit of verifying each input is 500s and 1000s for samples in MNIST and CIFAR-10 respectively.

| Dataset | $\epsilon_\infty$ | Method | Year | Natural Acc. | Certified Acc. |
|---|---|---|---|---|---|
| MNIST | 0.1 | COLT | 2020 | 99.20 | 97.10 |
| | | Crown-IBP | 2020 | 98.83 | 97.76 |
| | | IBP-Fast | 2021 | 98.84 | 97.95 |
| | | SABR | 2023 | **99.23** | 98.22 |
| | | Ours | - | 99.22 | **98.24** |
| | 0.3 | COLT | 2020 | 97.30 | 85.70 |
| | | Crown-IBP | 2020 | 98.18 | 92.98 |
| | | IBP-Fast | 2021 | 97.67 | 93.10 |
| | | SABR | 2023 | **98.75** | **93.40** |
| | | Ours | - | 98.64 | 93.22 |
| CIFAR-10 | 2/255 | COLT | 2020 | 78.40 | 60.50 |
| | | Crown-IBP | 2020 | 71.52 | 53.97 |
| | | IBP-Fast | 2021 | 66.84 | 52.85 |
| | | IBP-R | 2022 | 78.19 | 61.97 |
| | | SABR | 2023 | **79.24** | 62.84 |
| | | Ours | - | 78.75 | **62.99** |
| | 8/255 | COLT | 2020 | 51.70 | 27.50 |
| | | Crown-IBP | 2020 | 46.29 | 33.38 |
| | | IBP-Fast | 2021 | 48.94 | 34.97 |
| | | IBP-R | 2022 | 51.43 | 27.87 |
| | | SABR | 2023 | **52.38** | 35.13 |
| | | Ours | - | 51.18 | **35.44** |

certify. For the adversarial sample generation, we adopted PGD Madry et al. (2017) with 10 iterative steps, where the step size was set to 0.007. The total loss we use is $\mathcal{L} = \mathcal{L}_{sabr} + \alpha \sum_i \mathcal{L}_{var}^i + \beta \sum_i \mathcal{L}_{stb}^i$, where $\mathcal{L}_{sabr}$ is from Müller et al. (2022), $\alpha = 0.001$, and $\beta = 0.0005$. We used the Adam optimizer to update the network parameters with a batchsize of 256.

**Comparison with the State-of-the-arts**  We compare our method with existing certified training methods including COLT Balunović & Vechev (2020), CROWN-IBP Zhang et al. (2019b), IBP-Fast Shi et al. (2021), IBP-R De Palma et al. (2022), and SABR Müller et al. (2022) on different datasets with various perturbation magnitudes. Table 2 demonstrates the clean accuracy and the certified accuracy of baselines and our methods, where our method outperforms all baseline approaches in most experimental settings. The state-of-the-art method, SABR, propagates a small interval of the adversarial region to significantly reduce the over-approximation, while not considering neuron variance or neuron stability. In contrast, our method introduces variance loss and stability loss, respectively, for neuron variance reduction and neuron stability enhancement. As a result, our method outperforms state-of-the-art methods with 62.99% and 35.44% certified robustness on CIFAR-10 under the 2/255 and 8/255 $L_\infty$ perturbation. The advantages of our method are more obvious in the scenarios with higher perturbation magnitude because larger noise usually causes more unstable neurons, and explicitly reducing the neuron instability can benefit the certified robustness of the networks. We also outperform state-of-the-art methods with a 98.24% certified robustness on MNIST with the 0.1 $L_\infty$ perturbation. The performance enhancement in MNIST is not as significant as in CIFAR-10, since in MNIST, the samples are easier to be classified correctly and certified.

**SNR before and after training**  We show that the proposed losses effectively improve SNR and enhance neuron stability. Figure 7 shows the logarithm of the layer signal and the layer noise defined by equation 2 of different layers of CNN7 across all test samples in MNIST. Our model trained with the variance loss and the stability loss generally achieves higher signal and lower noise in all layers. We also show the line $S_i^2 = N_i^2$ in the figure. Samples located under the line have $S_i^2 > N_i^2$, indicating lower variance and fewer unstable neurons because the magnitude of the noise is smaller than that of the signal.

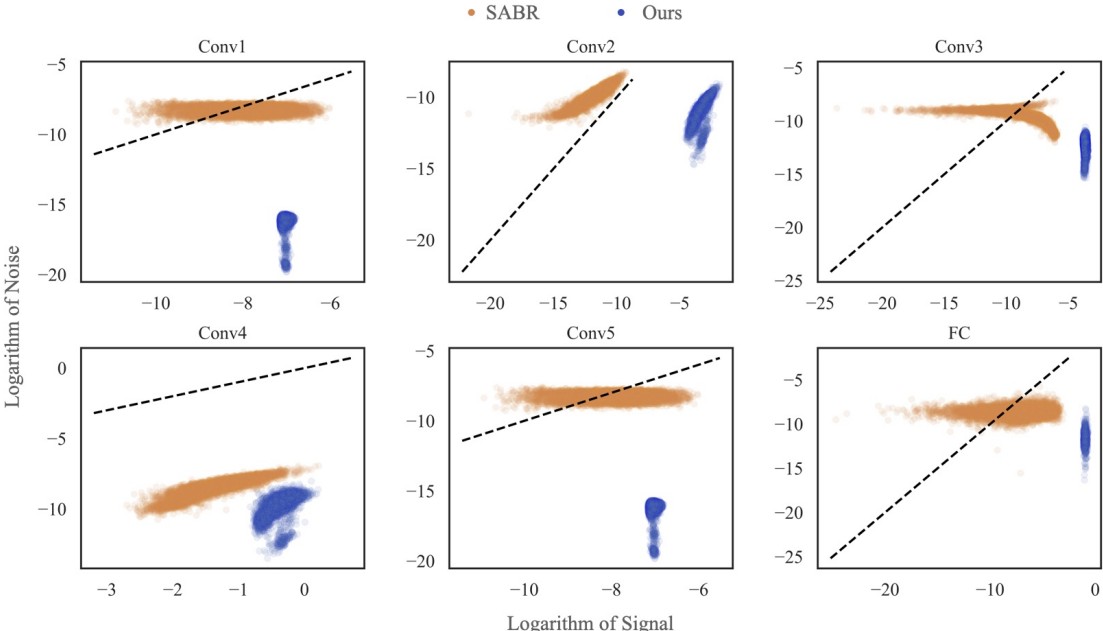

Figure 7: $\log S_i^2$ and $\log N_i^2$ across different layers of CNN7 for 10000 test samples. Our method achieves higher signal and lower noise than the baseline. The dashed line represents $S_i^2 = N_i^2$. Samples above the dashed line generally have more unstable neurons because their noise is larger than signal.

An ablation study of $\alpha$ and $\beta$ is shown in table 3. The results report clean accuracy and certified accuracy on CIFAR-10 under 2/255 perturbations. We can see that the performance does not change monotonously with either parameter. We hypothesize that these two losses are coupled with each other, therefore leading to this non-monotonous behavior. We will investigate a more decoupled design to isolate the effect of the loss in the future. We will also investigate how to choose the best alpha and beta under the current design in future work.

Table 3: Comparison of $\alpha$ and $\beta$ values on CIFAR-10 under 2/255 perturbations. Numbers denote natural accuracy and certified accuracy respectively.

|  | $\beta = 0.0001$ | $\beta = 0.0005$ | $\beta = 0.001$ |
|---|---|---|---|
| $\alpha = 0.0001$ | 79.00%/62.80% | 78.53%/62.01% | 79.05%/61.97% |
| $\alpha = 0.001$ | 78.52%/62.75% | 78.75%/62.99% | 77.13%/62.68% |
| $\alpha = 0.01$ | 78.23%/62.31% | 77.82%/62.10% | 77.69%/62.90% |

## 5.2 Adversarial training

**Neuron variance and adversarial robustness** We have shown that the SNR losses improve certified robustness. Next, we show that the adversarial robustness is also related to neuron variance and SNR, and can be improved by the variance loss.

We draw the SNR curves for models trained adversarially versus those trained with standard methods, as sourced from RobustBench Croce et al. (2020), to show the correlation between neuron variance and adversarial robustness. We consider the three most popular adversarial learning architectures Croce et al. (2020): WideResNet-28-10, WideResNet-34-20, and WideResNet-70-16. As shown in fig. 8, adversarially trained models have significantly higher SNRs than standard trained models. The denoising behavior is consistent across all models. Very interestingly, we find that the signal processing can be roughly divided into three stages, which we call *pit, platform, and incline*, exactly corresponding to the three block groups

of the ResNet architecture. The pattern is consistent with what Allen-Zhu & Li (2022) have proposed: adversarially trained models learn to purify spurious features while standard trained models do not.

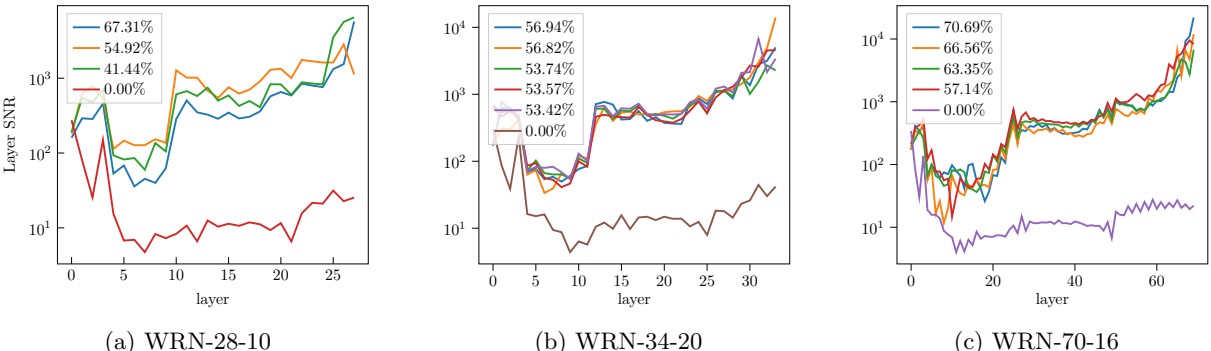

(a) WRN-28-10                    (b) WRN-34-20                    (c) WRN-70-16

Figure 8: SNR curves of models at different robustness levels from RobustBench. The number in the legend indicates the adversarial robustness of the model. Robust models generally have a high and increasing SNR curve. The non-robust model (0.00%) shows a low SNR curve with a different trend.

**Variance loss improves adversarial robustness** We next show that the adversarial robustness can be improved by reducing neuron variance. We combine the variance loss with two popular adversarial training methods, AD Madry et al. (2017) and TRADES Zhang et al. (2019a), by two hyperparameters $\gamma \mathcal{L}_{trades} + w \mathcal{L}_{var}$ and $\gamma \mathcal{L}_{ad} + w \mathcal{L}_{var}$. Following the setup described in TRADES Zhang et al. (2019a), a CNN architecture with two convolutional layers and two fully-connected layers is employed for MNIST. Settings include perturbation $\epsilon = 0.1$, perturbation step size $\eta = 0.01$, number of iterations $K = 20$, learning rate $lr = 0.01$, batch size $m = 128$, and training epoch is 50. For CIFAR-10, ResNet-18 serves as the classification. Here, perturbation $\epsilon = 8/255$, perturbation step size $\eta = 0.007$, number of iterations $K = 10$, learning rate $lr = 0.1$, batch size $m = 128$, and training epoch is 100.

The trade-off curve between natural accuracy and robust accuracy is illustrated, highlighting that variance loss results in higher robustness under the same natural accuracy. We plot the curve by fix the value of $\gamma$ in [0.1, 0.2, 0.4, 0.6, 0.8, 1, 2, 3, 4, 5]. When the value of $\gamma$ is fixed, we modified the value of $w$ for many times. For each value of $\gamma$, the corresponding experimental results of $w$ with the best performance are selected and connected as lines, and the other experiments are drawn as scatter points. As depicted in fig. 9, TRADES+SNR exhibits a more favorable trade-off than TRADES alone. Interestingly, AD, previously considered inferior to TRADES Zhang et al. (2019a), surpasses TRADES when augmented with the variance loss. The curve of TRADES is based on the original results from the paper Zhang et al. (2019a). These findings affirm the positive impact of reducing neuron variance on adversarial robustness.

**Decomposition of certified training** Certified training methods usually directly optimize the over-approximated set (Mao et al., 2023b). However, with the variance loss and stability loss, we essentially decompose certified training into two parts: 1. improving the robustness of the worst-case output (true robustness) by reducing neuron variance; 2. reducing over-approximation (the gap between true robustness and certified robustness) by minimizing unstable neurons and neuron variance. For part 1, although the worst-case output is difficult to compute exactly, adversarial attacks can find close approximations of it, making adversarial robustness a good proxy for true robustness. Therefore, the experiment results on adversarial robustness confirm the correlation between neuron variance and the true robustness. For part 2, we have shown that neuron variance and neuron stability are related to over-approximation in section 3.

## 6 Conclusion

This study addresses the critical challenge of over-regularization in certified training of neural networks. By introducing and analyzing the novel concepts of neuron variance and neuron stability, we have uncovered

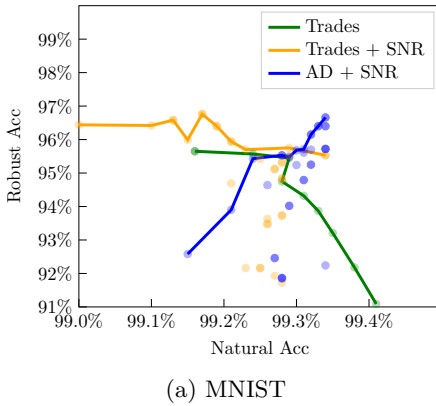
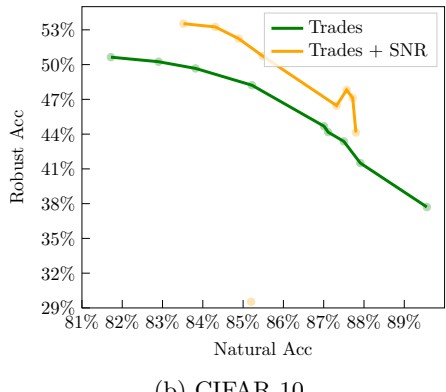

(a) MNIST

(b) CIFAR-10

Figure 9: Trade-off between natural accuracy and robust accuracy. We test on multiple hyper-parameters and draw the upper curve the points as the trade-off curve. In general, comparing to TRADES, our method achieves a better trade off between robust accuracy and natural accuracy.

their significant roles in influencing the model's certified robustness. Our pioneering extension of the Signal-to-Noise Ratio (SNR) into the domain of model robustness has yielded a novel analytical perspective, leading to the development of SNR-inspired losses specifically designed to optimize neuron variance and stability. Empirical and theoretical analyses underscore the effectiveness of our SNR-based losses in improving certified robustness, demonstrably outperforming existing methods on benchmark datasets such as MNIST and CIFAR-10. Furthermore, our exploration into adversarial training highlights a positive correlation between neuron variance and the true robustness of a model. By optimizing neuron variance, our approach achieves a superior trade-off between standard accuracy and robust accuracy compared to baseline methods.

One limitation of our method is the added complexity in training dynamics due to the application of stability and activation losses across multiple layers. This complexity necessitates a careful balance between these new losses and the original loss functions, introducing challenges in achieving optimal training outcomes. For future work, we aim to delve deeper into this issue, exploring strategies to streamline the balancing process and enhance the efficacy of the training regimen.

# 7    Acknowledgements

This material is based upon work supported by The Boeing Company. Any opinions, findings, and conclusions or recommendations expressed in this material are those of the authors and do not necessarily reflect the views of The Boeing Company.

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

## A  Appendix

### A.1  2 unstable neurons

In this case, we have $c_x - r < 0 < c_x + r$ and $c_y - r < 0 < c_y + r$. Without loss of generality, we assume $c_y > c_x$.

**Ground truth**   The ground truth $Y$-radius is $\frac{\sqrt{2}}{2} \max\{c_y + r, c_x + c_y + r\} = \frac{\sqrt{2}}{2}(c_y + r + \max\{0, c_x\})$.

**Box**   over-approximate the non-convex shape ReLU$(B)$ with a hyperrectangle $\{(x, y) \mid x \in [0, c_x + r], y \in [0, c_y + r]\}$. The after $W_2$ rotation $Y$-radius is: $(c_x + r)\sin\theta + (c_y + r)\cos\theta = \frac{\sqrt{2}}{2}(c_x + c_y + 2r)$.

**Star**   representation of the $L_1$-norm ball is

$$(c_x, c_y) + \alpha_1(r, 0) + \alpha_2(0, r) \tag{5}$$

$$\text{s.t.} \begin{bmatrix} 1 & 1 \\ 1 & -1 \\ -1 & 1 \\ -1 & -1 \end{bmatrix} \cdot \begin{bmatrix} \alpha_1 \\ \alpha_2 \end{bmatrix} < \begin{bmatrix} 1 \\ 1 \\ 1 \\ 1 \end{bmatrix} \tag{6}$$

$$\tag{7}$$

After ReLU activation, the star over-approximated reachable set is

$$(0,0) + \alpha_1(0,0) + \alpha_2(0,0) + \beta_1(1,0) + \beta_2(0,1) \tag{8}$$

$$\text{s.t.} \begin{bmatrix} 1 & 1 & 0 & 0 \\ 1 & -1 & 0 & 0 \\ -1 & 1 & 0 & 0 \\ -1 & -1 & 0 & 0 \\ 0 & 0 & -1 & 0 \\ r & 0 & -1 & 0 \\ -u_x/2 & 0 & 1 & 0 \\ 0 & 0 & 0 & -1 \\ 0 & r & 0 & -1 \\ 0 & -u_y/2 & 0 & 1 \end{bmatrix} \cdot \begin{bmatrix} \alpha_1 \\ \alpha_2 \\ \beta_1 \\ \beta_2 \end{bmatrix} < \begin{bmatrix} 1 \\ 1 \\ 1 \\ 1 \\ 0 \\ -cx \\ u_x/2 \\ 0 \\ -c_y \\ u_y/2 \end{bmatrix} \tag{9}$$

$$\tag{10}$$

The after rotation $Y$-radius is $\beta_1 \sin\theta + \beta_2 \cos\theta$, where $(\beta_1, \beta_2)$ must be a vertex of the feasible region of Star. We enumerating all possible vertices of the feasible region and find that the after-rotation $Y$-radius only depends on the max of two vertices $A$ and $B$'s y-value, where $A$ and $B$ are derived from

$$A: \begin{cases} \alpha_1 + \alpha_2 & = 1 \\ -\alpha_1 + \alpha_2 & = 1 \\ -u_x/2 \cdot \alpha_1 + \beta_1 & = u_x/2 \\ r\alpha_2 - \beta_2 & = -c_y \end{cases} \quad B: \begin{cases} \alpha_1 + \alpha_2 & = 1 \\ \alpha_1 - \alpha_2 & = 1 \\ r\alpha_1 - \beta_1 & = -c_x \\ -u_y/2 \cdot \alpha_2 + \beta_2 & = u_y/2 \end{cases} \tag{11}$$

$$\tag{12}$$

$$A: \begin{cases} \alpha_1 & = 0 \\ \alpha_2 & = 1 \\ \beta_1 & = (c_x + r)/2 \\ \beta_2 & = c_y + r \end{cases} \quad B: \begin{cases} \alpha_1 & = 1 \\ \alpha_2 & = 0 \\ \beta_1 & = c_x + r \\ \beta_2 & = (c_y + r)/2 \end{cases} \tag{13}$$

$$\tag{14}$$

Because we assumed $c_y > c_x$, The after rotation $Y$-radius is $\max\{\frac{\sqrt{2}}{2}(\beta_1 + \beta_2)\} = \frac{\sqrt{2}}{2}(\frac{1}{2}c_x + c_y + \frac{3}{2}r)$.

### A.2  1 unstable neuron

Without loss of generality, we assume $l_x < 0 < u_x$ and $y$-axis is stable. There are two cases: 1) $l_y < u_y \leq 0$ and 2) $0 \leq l_y < u_y$. In the first case, all methods give the results because it degenerates to a line segment. We consider the second case.

**Ground truth**  The ground truth $Y$-radius is $\sqrt{2}r$.

**Box**  over-approximates the non-convex shape ReLU($B$) with a hyperrectangle $\{(x, y) \mid x \in [0, c_x + r], y \in [c_y - r, c_y + r]\}$. The after $W_2$ rotation $Y$-radius is: $(c_x + r)\sin\theta + 2r\cos\theta = \frac{\sqrt{2}}{2}(c_x + 3r)$.

**Star** After ReLU activation, the star over-approximated reachable set is

$$(0, c_y) + \beta_1(1, 0) + \alpha_2(0, r) \tag{15}$$

$$\text{s.t.} \begin{bmatrix} 1 & 1 & 0 \\ 1 & -1 & 0 \\ -1 & 1 & 0 \\ -1 & -1 & 0 \\ 0 & 0 & -1 \\ r & 0 & -1 \\ -u_x/2 & 0 & 1 \end{bmatrix} \cdot \begin{bmatrix} \alpha_1 \\ \alpha_2 \\ \beta_1 \end{bmatrix} < \begin{bmatrix} 1 \\ 1 \\ 1 \\ 1 \\ 0 \\ -cx \\ u_x/2 \end{bmatrix} \tag{16}$$

$$\tag{17}$$

The after rotation $Y$-max is $\beta_1 \sin\theta + (c_y + r\alpha_2)\cos\theta$, where $(\beta_1, \alpha_2)$ must be a vertex of the feasible region of Star. We enumerate all possible vertices of the feasible region and find that the after-rotation $Y$-max only depends on the max of two vertices:

$$\begin{cases} \alpha_1 + \alpha_2 & = 1 \\ -\alpha_1 + \alpha_2 & = 1 \\ -u_x/2 \cdot \alpha_1 + \beta_1 & = u_x/2 \\ r\alpha_2 - \beta_2 & = -c_y \end{cases} \qquad \begin{cases} \alpha_1 + \alpha_2 & = 1 \\ \alpha_1 - \alpha_2 & = 1 \\ r\alpha_1 - \beta_1 & = -c_x \\ -u_y/2 \cdot \alpha_2 + \beta_2 & = u_y/2 \end{cases} \tag{18}$$

$$\tag{19}$$

$$\begin{cases} \alpha_1 & = 0 \\ \alpha_2 & = 1 \\ \beta_1 & = (c_x + r)/2 \end{cases} \qquad \begin{cases} \alpha_1 & = 1 \\ \alpha_2 & = 0 \\ \beta_1 & = c_x + r \end{cases} \tag{20}$$

$$\tag{21}$$

That is, $((c_x + r)/2, c_y + r)$ and $(c_x + r, c_y)$. Because $c_x - r < 0$, the after $\pi/4$ rotation y-max is

$$\frac{\sqrt{2}}{2} \max\{\frac{1}{2}c_x + c_y + \frac{3}{2}r, c_x + c_y + r\} = \frac{\sqrt{2}}{2}(\frac{1}{2}c_x + c_y + \frac{3}{2}r) \tag{22}$$

Similarly, the after rotation $Y$-min depends on

$$\begin{cases} \alpha_1 + \alpha_2 & = 1 \\ -\alpha_1 + \alpha_2 & = 1 \\ -u_x/2 \cdot \alpha_1 + \beta_1 & = u_x/2 \\ r\alpha_2 - \beta_2 & = -c_y \end{cases} \qquad \begin{cases} \alpha_1 + \alpha_2 & = 1 \\ \alpha_1 - \alpha_2 & = 1 \\ r\alpha_1 - \beta_1 & = -c_x \\ -u_y/2 \cdot \alpha_2 + \beta_2 & = u_y/2 \end{cases} \tag{23}$$

$$\tag{24}$$

$$\begin{cases} \alpha_1 & = 0 \\ \alpha_2 & = -1 \\ \beta_1 & = \max\{c_x, 0\} \end{cases} \qquad \begin{cases} \alpha_1 & = -\max\{c_x, 0\}/r \\ \alpha_2 & = \max\{c_x, 0\}/r - 1 \\ \beta_1 & = 0 \end{cases} \tag{25}$$

$$\tag{26}$$

That is, $(\max\{c_x, 0\}, c_y - r)$ and $(0, c_y - r + \max\{c_x, 0\})$. The after $\pi/4$ rotation y-min is

$$\frac{\sqrt{2}}{2}(c_y - r + \max\{c_x, 0\}). \tag{27}$$

The after rotation $Y$-radius is

$$\frac{\sqrt{2}}{2}(\frac{1}{2}c_x + c_y + \frac{3}{2}r) - \frac{\sqrt{2}}{2}(c_y - r + \max\{c_x, 0\}) = \frac{\sqrt{2}}{2}(\frac{1}{2}c_x + \frac{5}{2}r - \max\{c_x, 0\}) \tag{28}$$

### A.3   0 unstable neurons

There are two cases: 1) $c_x + r < 0$ and $c_y + r < 0$; 2) $0 < c_x - r$ and $0 < c_y - r$. In the first case, all methods give an empty set. We consider the second case.

**Ground truth**   The ground truth $Y$-radius is $\sqrt{2}r$.

**Box**   over-approximate the ReLU$(B)$ with a hyperrectangle $\{(x, y) \mid x \in [c_x - r, c_x + r], y \in [c_y - r, c_y + r]\}$. The after $W_2$ rotation $Y$-radius is $2r \sin \theta + 2r \cos \theta = 2\sqrt{2}r$

**Star**   has no over-approximation when there is no unstable neurons, therefore it has the same radius as the ground truth $\sqrt{2}r$.

