# OpenReview forum: "Improve Certified Training with Signal-to-Noise Ratio Loss to Decrease Neuron Variance and Increase Neuron Stability"
_TMLR — Accepted by TMLR_

### Review · Reviewer_kTim · 2024-04-02

**Summary Of Contributions:**

The paper investigates the impact of neuron stability on cerfitied adversarial robustness. The connection between neuron stability and tightness of approximation bound of several cerfitication methods is first empirically studied and then analyzed mathematically. The Signal-to-Noise Ratio (SNR) loss is proposed to induce more neuron stability and improve cerfitied adversarial robustness. The experiment on MNIST and CIFAR10 is shown to validate the effectiveness of the proposed SNR loss.

**Audience:**

Yes

**Broader Impact Concerns:**

Not applicable.

**Claims And Evidence:**

No

**Requested Changes:**

From Fig. 8, it is hard to see the correlation between adversarial robustness and Layer SNR. Please give a quantitative result to show this correlation.

**Strengths And Weaknesses:**

Strengths:


The paper builds a connection between neuron stability and tightness of adversarial robustness certification approaches with interesting empirical evidence.

The paper writing is good-quality in general.


Weaknesses:

The major weakness is that the proposed SNR loss is not quite effective at improving cerfitied accuracy according to Table 2. For instance, SNR is better than SABR with a margin of 0.15% when $\epsilon$=2/255 and 0.3% when $\epsilon$=8/255. This weakness is the reason why I choose No in the Claims and Evidence section.

From Fig. 8, it is hard to see the correlation between adversarial robustness and Layer SNR.

---

> ### Author Response · Authors · 2024-04-11
> **Response to reviewer kTim**
>
> Dear reviewer, we greatly appreciate your valuable time and opinion. Please find our response to your concerns and suggestions below.
>
> Q: The proposed SNR loss is not quite effective at improving certified accuracy according to Table 2. SNR is better than SABR with a margin of 0.15% when eps=2/255 and 0.3% when eps=8/255.
>
> A: The advances in certified robustness often come as a marginal improvement because improving certified robustness is significantly harder than improving nominal accuracy. For example, SABR [1], published at ICLR 2023, is better than its prior SOTA by 0.87% when eps=2/255 and 0.16% when eps=8/255.  A recent method, STAPS [2], published at NeurIPS 2023, is better than its prior SOTA (SABR) by 0.14% when eps=2/255 and -0.48% when eps=8/255. Our method shows a better improvement than STAPS in both cases. Furthermore, with the development of certified training techniques, the improvement has become more and more difficult. Therefore, we believe our approach shows a meaningful improvement in the context of certified robustness.
> [1] SBAR: https://openreview.net/forum?id=7oFuxtJtUMH
> [2] STAPS: https://openreview.net/forum?id=T2lM4ohRwb
>
> Q: From Fig. 8, it is hard to see the correlation between adversarial robustness and Layer SNR.
>
> A: Figure 8 shows that adversarially trained models have a different SNR curve than nominally trained models, which reveals a general correlation between adversarial training and SNR. However, we do not observe a specific correlation between layer SNR and adversarial robustness. We hypothesize that this is caused by the nature of adversarial training, which only focuses on the robustness of the last layer output. Adversarial training does not care about the robustness of intermediate layer output. Therefore, the intermediate layer SNR didn’t increase as the adversarial robustness increased. We will look into this problem in the future and design experiments to verify this hypothesis, trying our best to give a quantitative relationship.
>
> Please let us know if you have any further concerns. Thank you!

---

> > ### Comment · Reviewer_kTim · 2024-05-09
> > **Follow-up**
> >
> > Thanks for the response. I checked the STAPS paper, where the proposed method achieves state-of-the-art performance on Tiny-ImageNet.
> >
> > The argument that adversarial training only focuses on the robustness of the last layer output and does not care about the robustness of intermediate layer output sounds weird to me, as it is not feasible to have vulnerable intermediate layers but a robust final layer.

---

> ### Author Response · Authors · 2024-04-30
>
> Dear Reviewer,
>
> As the rebuttal deadline approaches, please let us know if you have any further concerns. We aim to ensure your satisfaction. Thank you!

---

> ### Author Response · Authors · 2024-05-10
> **Response to follow-up**
>
> Dear reviewer,
>
> Thank you for your feedback. We didn't conduct tests on Tiny-ImageNet because the training and verification are taking too long. It takes more than 7 days to train the model with certification for a single run on our server. We will investigate this dataset in the future when we have more computational resources.
>
> To clarify the behavior of intermediate layers, let us consider a straightforward example:
>
> Imagine a simple 1-2-1 network, h = ReLU(W1 * x + b1), y = W2 * h. Here, W1 = [10, 5], b1 = [-10,-10], W2 = [1; -2]. Observing the network's response to various inputs:
>
> x = [1]:  h = [0, 0],   y = [0].
>
> x = [2]:  h = [10, 0],  y = [10].
>
> x = [3]:  h = [20, 5],  y = [10].
>
> x = [4]:  h = [30, 10], y = [10].
>
> The output is stable around x = 3, demonstrating robustness in the final layer. However, the hidden layer changes a lot near x = 3, indicating vulnerability. The reason is that later layers can counterbalance the effects of earlier layers' fluctuations.
>
> We appreciate your insights and look forward to further discussions.

---

### Review · Reviewer_Vhde · 2024-04-06

**Summary Of Contributions:**

The paper proposes to introduce Signal-Noise-Ratio (SNR) to improve the performance of adversarial robustness and certified robustness caused by over-approximation problem . Specifically, the paper first starts with analyzing the neuron variance and neuron stability to find their correlation with over-approximation. Through a case study with numerical analysis, it finds the reachable sets is sensitive to the change of the neuron status but the exact reachable set is insensitive. It also tries to theoretically analyze the radius of the reachable sets with a two layer neural networks. Inspired by the findings, the paper then proposes layer SNR loss to penalize the neuron variance and Neuron SNR to reduce the number of unstable neurons. Experiments conducted in MNIST and CIFAR10 both on certified training and adversarial training shows the proposed method could achieve a better robustness than other baselines.

**Audience:**

Yes

**Broader Impact Concerns:**

No ethic concerns.

**Claims And Evidence:**

No

**Requested Changes:**

Please see the Cons and Minors. In sum,
1. Definition and more introduction on the reachable sets.
2. Evidence on Neuron SNR loss to reduce # of unstable neurons.
3. Experiments on different $\alpha$ and $\beta$.
4. Stronger baselines in adversarial training setting.

**Strengths And Weaknesses:**

Pros:
1. The paper is in general well-written and easy to follow. There are some typos and improvements I find as listed in the Minors.
2. The paper proposes a new perspective to analyze the over-approximation problem with neuron variance and neuron stability, which is interesting and novel.
3. The proposed regularizer could be used both in adversarial training and certified training and shows some improvements.

Cons:
1. There is not enough introduction on how to get the reachable set and the definition of the radius of the reachable sets. As the most essential analytic tools for the hypothesis, it is necessary to illustrate them very clearly.
2. The Neuron SNR loss's motivation is unclear to me to reduce the number of unstable neurons. More evidence is needed.
3. The experiments needs a detailed analysis on the choice of $\alpha$ and $\beta$ especially under certification case. There are only one set of $\alpha$ and $\beta$ tested and it is unclear whether the proposed method could still achieve good results under different $\alpha$ and $\beta$.
4. There is no ablation experiments on separating the two losses.
5. The experiments on the adversarial robustness is weak with only comparing with two old baselines AD and TRADES. Stronger defense in the RobustBench should be considered.

Minors:
1. No axis labels in Figure 3 and 6.
2. table 1 should be "Table 1" in Page 7.
3. $x_k$ where k should be defined from 1 to m in Page 8.
4. fi(x_0) in Page 8 should be in latex symbol.
5. True robustness in Page 11 should be somehow defined.

---

> ### Author Response · Authors · 2024-04-11
> **Response to reviewer Vhde**
>
> Dear reviewer, we greatly appreciate your valuable time and opinion. Please find our response to your concerns and suggestions below.
>
> Q. Definition and more introduction on the reachable sets.
>
> A: Thanks for the suggestion. Briefly, the reachable sets—namely Box, Zonotope, and Star—are computed using methods from prior research. Box representation keeps a dimension-wise lower bound and upper bound. Zonotope is defined as {x | x = c + \sum_i a_i g_i, s.t. |a|<1}, with c and g_i as layer-output-dimensional vectors. c denotes the center of the set, g_i is a generator of the set, and a is a scalar. In short, Zonotope contains all points that can be represented as the center plus a weighted sum of the generators, where the L1 norm of the weight should be less than 1. Star representation uses a similar representation as Zonotope but with a different constraint on the weight: {x | x = c + \sum_i a_i g_i, s.t. A a < b}. This A a < b is more flexible than |a| < 1, therefore Star is tighter than Zonotope but is slower to compute. We will include more introductions on the reachable sets in the final revision.
>
>
> Q: Motivation and evidence on Neuron SNR loss to reduce # of unstable neurons.
>
> A: 1. The motivation of the neuron SNR is that a smaller noise range increases the probability that the noise range does not cross zero. Consider a simple example where we randomly choose an interval from -10 to 10. We call the interval unstable if it contains zero. If the length of the interval is 11,  then it is 100% to be unstable. However, if the length of the interval is 1, then it only has a 5% probability of being unstable. Therefore, we want to use Neuron SNR loss to reduce the noise range, thereby making it more likely to be stable.
> 2. The evidence that Neuron SNR reduces the number of unstable neurons is shown in Figure 7. We plotted the signal range and noise range for 10000 test samples. Each dot in the figure represents a test sample. The signal represents the distance of an interval to zero, and the noise represents the length of the interval. When noise is larger than signal, that means the interval contains zero, and is an unstable neuron. We plot a dashed line in the figure that represents when the signal scale equals the noise scale. Dots above this line are unstable neurons, and dots below this line are stable neurons. Compared to the baseline, our method makes more points below this line, making them stable neurons. We will add more analysis for this figure to make the conclusion more clear.
>
>
> Q: Experiments on different alpha and beta.
>
> A: We agree an ablation study on different alpha and beta is important to understand the underlying mechanism of the method. We tried to do some preliminary tests of different alpha and beta during exploration. However, certified training and the following verification are extremely time-consuming compared to nominal neural network training. It takes 2-3 days to finish one experiment with a set of alpha and beta. Therefore, we did not provide a thorough study of different combinations of alpha and beta in the first draft. Currently, we are running more experiments to study the effect of different alpha and beta. We will share with you the ablation results as soon as we finish these experiments.
>
>
> Q: Stronger baselines in adversarial training settings.
>
> A: We compare with AD and TRADES because they fall in the same category of our method: regularization loss from human priors. In contrast, recent advances in adversarial training heavily rely on data augmentation, which falls into a different category. All the leading methods on RobustBench use synthesized data or external data to augment the model. The usage of external data introduces more implicit priors of real-world images, making the model more robust. Our method does not show significant improvement when the data is sufficient, because all the priors have already been encoded in the data. Human priors are no longer necessary. However, when the data is insufficient, our method introduces a better prior than the de facto methods, AD and TRADES. This is the major motivation for comparing our method with AD and TRADES.
>
>
> We will revise the typos in the final version. Please let us know if you have any further concerns. Thank you!

---

> > ### Author Response · Authors · 2024-04-18
> > **Ablation study results**
> >
> > Dear reviewer,
> >
> > We have conducted the ablation study. The results report clean accuracy and certified accuracy on Cifar-10 under 2/255 perturbation. We can see that the performance does not change monotonously with either parameter. We hypothesize that these two losses are coupled with each other, therefore leading to this non-monotonous behavior. We will investigate a more decoupled design to isolate the effect of the loss in the future. We will also investigate how to choose the best alpha and beta under the current design in future work.
> >
> > |    |  beta=0.0001 |	beta=0.0005  |	beta=0.001|
> > | -------- | ------- | ------- | ------- |
> > | alpha=0.0001	| 79.00/62.80 |	78.53/62.01 |	79.05/61.97|
> > | alpha=0.001	| 78.52/62.75 | 	78.75/62.99 | 	77.13/62.68|
> > | alpha=0.01	| 78.23/62.31 | 	77.82/62.10 | 	77.69/62.90|

---

> > ### Author Response · Authors · 2024-04-30
> >
> > Dear Reviewer,
> >
> > As the rebuttal deadline approaches, please let us know if you have any further concerns. We aim to ensure your satisfaction. Thank you!

---

### Review · Reviewer_W1rM · 2024-04-09

**Summary Of Contributions:**

This study addresses the challenge of over-regularization in certified training, which tends to compromise the certified robustness of models. It introduces and investigates the roles of neuron variance and neuron stability in fostering over-regularization and their effects on a model's certified robustness. Adapting the concept of Signal-to-Noise Ratio (SNR) to the domain of model robustness, the research provides a fresh perspective and devises SNR-inspired loss functions aimed at optimizing neuron variance and stability to alleviate over-regularization. Validated through both empirical and theoretical analyses, this SNR-based methodology demonstrates enhanced performance over existing approaches on the MNIST and CIFAR-10 datasets. Furthermore, the study explores the relationship between neuron variance and adversarial robustness within the context of adversarial training. This exploration identifies a positive correlation that leads to an optimized balance between standard accuracy and robust accuracy, thereby exceeding the efficacy of baseline methods.

**Audience:**

Yes

**Broader Impact Concerns:**

No concerns for this section.

**Claims And Evidence:**

Yes

**Requested Changes:**

1. It would be great to add some simple theoretical analysis using a simple network, showing how the neuron variance and stability affect the certified robustness.

**Strengths And Weaknesses:**

Strengths:

1. This paper considers neuron variance and stability that can be used to see how a network is regulated, which is interesting.

2. Based on the above motivation, SNR is used to decrease the variance and increase the stability, to reach a better certified network.

3. Some experiments are provided to see if the proposed idea can work in some constructed datasets.

4. Experiments are solid and conducted on many models.

Weaknesses:

1. It would be great to add some simple theoretical analysis using a simple network, showing how the neuron variance and stability affect the certified robustness.

2. In Table 2, it should be \epsilon_{\rm inf}

---

> ### Author Response · Authors · 2024-04-11
> **Response to reviewer W1rM**
>
> Dear reviewer, we greatly appreciate your valuable time and opinion. Please find our response to your concerns and suggestions below.
>
> Q: Theoretical analysis using a simple neural network.
>
> A: We appreciate your feedback. In Section 3.3, we have included an analytical study of a two-layer neural network with two-dimensional input and output. We examined how the over-approximation is influenced by the number of unstable neurons and the output range of neurons, reflecting the impacts on certified robustness due to neuron stability and variance. The detailed derivation is attached in the appendix. We will clarify these connections further in the revised manuscript. Thank you.
>
> We will also revise the typo in the final version. Please let us know if you have any further concerns. Thank you!

---

> ### Author Response · Authors · 2024-04-30
>
> Dear Reviewer,
>
> As the rebuttal deadline approaches, please let us know if you have any further concerns. We aim to ensure your satisfaction. Thank you!

---

### Comment · Action_Editor_1ZYz · 2024-03-13
**Message from AE**

Hi reviewers Vhde,

This is a second reminder. If you have noticed about the assignment of this submission to you, please first of all acknowledge that you are aware of your reviewing task. Thank you!

Best,
AE

---

### Comment · Action_Editor_1ZYz · 2024-04-04

Hi reviewers xdTf, Vhde, and W1rM,

I understand your are busy, but please submit your late review for this submission at your earliest convenience. Thank you!

AE

---

> ### Comment · Action_Editor_1ZYz · 2024-04-08
>
> Dear reviewers xdTf and W1rM,
>
> Please let me know whether you can receive my messages sent through openreview by replying to this post. Thank you.
>
> AE

---

### Decision · Action_Editor_1ZYz · 2024-05-17

**Recommendation:** Accept with minor revision

**Comment:**

This paper studied certified robustness and proposed a signal-to-noise ratio loss to improve this robustness. It has nice results both theoretical and empirical. After the rebuttal, two reviewers voted for acceptance and one for rejection --- the rejection argument was as follows:
> (recommendation) The effectiveness of the proposed method is not supported by the empirical evidence

> (review) The major weakness is that the proposed SNR loss is not quite effective at improving cerfitied accuracy according to Table 2. For instance, SNR is better than SABR with a margin of 0.15% when $\epsilon$=2/255 and 0.3% when $\epsilon$=8/255. This weakness is the reason why I choose No in the Claims and Evidence section.

However, I think it is fine, since certified robustness is much harder than empirical robustness, as the authors pointed out in their rebuttal. Even if the improvements look minor to some researchers, some other researchers may find this paper interesting and useful. Therefore, I would like to accept this submission.

One point I want to mention is that your abstract is really too short! It explains the title --- nothing more than that. A good abstract should have both the motivation part and the contribution part, so please consider to add the motivation part for your abstract.

**Audience:**

Yes.

**Claims And Evidence:**

Yes, I personally think so.

---

> ### Author Response · Authors · 2024-05-19
> **Reply to the decision**
>
> We greatly appreciate the effort and time of the Action Editor and all reviewers! We will revise the paper based on the feedback. Thank you all again!